evolution, ecology, genetics

hybridization, repeated adaptive radiation, adaptive radiation, speciation, evolvability, individual-based model

**Author for correspondence:**
Kotaro Kagawa
e-mail: kagawakoutarou@gmail.com

### PUBLISHING

# The propagation of admixture-derived adaptive radiation potential

Kotaro Kagawa[1,2,3] and Ole Seehausen[1,2]

[1]Department of Fish Ecology and Evolution, Center for Ecology, Evolution and Biogeochemistry, Swiss Federal Institute of Aquatic Science and Technology, Seestrasse 79, 6047 Kastanienbaum, Switzerland
[2]Aquatic Ecology and Evolution, Institute of Ecology and Evolution, University of Bern, Hochschulstrasse 6, 3012 Bern, Switzerland
[3]Graduate School of Life Sciences, Tohoku University, 2-1-1 Katahira, Aoba, Stendai, Miyagi 980-8578, Japan

KK, 0000-0002-5749-5679; OS, 0000-0001-6598-1434

Adaptive radiations (ARs) frequently show remarkable repeatability where single lineages undergo multiple independent episodes of AR in distant places and long-separate time points. Genetic variation generated through hybridization between distantly related lineages can promote AR. This mechanism, however, requires rare coincidence in space and time between a hybridization event and opening of ecological opportunity, because hybridization generates large genetic variation only locally and it will persist only for a short period. Hence, hybridization seems unlikely to explain recurrent AR in the same lineage. Contrary to these expectations, our evolutionary computer simulations demonstrate that admixture variation can geographically spread and persist for long periods if the hybrid population becomes separated into isolated sub-lineages. Subsequent secondary hybridization of some of these can reestablish genetic polymorphisms from the ancestral hybridization in places far from the birthplace of the hybrid clade and long after the ancestral hybridization event. Consequently, simulations revealed conditions where exceptional genetic variation, once generated through a rare hybridization event, can facilitate multiple ARs exploiting ecological opportunities available at distant points in time and space.

## 1. Background

Adaptive radiation (AR) is a major focus in evolutionary biology. One remarkable observation in empirical AR research is that the same lineage often made multiple ARs at geographically distant places or recurrent radiations in the same site through time, while closely related and ecologically similar lineages fail to radiate [1–9]. Why and how do certain lineages repeatedly exhibit such extraordinary evolvability?

Growing empirical and theoretical evidence suggest that hybridization between genetically distant lineages can promote AR by assembling genes from distinct parental lineages into a wide array of novel genotypes [10–19] (the hybrid swarm origin hypothesis of AR [20]). However, at first consideration, such a contingency-dependent mechanism may seem unlikely to explain the repeated occurrence of AR within a lineage. While only hybridization between sufficiently distant evolutionary lineages can generate much genotypic novelty, they must not be too distant to produce offspring of significant intrinsic fitness [14]. Such hybridization is probably rare and its coincidence in space and time with opening up of ecological opportunity is even rarer. It seems extremely unlikely that the right constellation of circumstances comes together in the same lineage repeatedly to explain recurrent AR. However, recent empirical studies suggest another possibility: genetic variation once generated by a singular hybridization event may permanently increase evolvability of the entire hybrid clade, facilitating recurrent AR in the clade. Especially, genomic studies of African cichlid fishes suggest that genetic variation generated through a single

ancient hybridization episode has promoted several ARs in geographically distant lakes at different points in time in eastern central Africa [12]. This may imply that the lineage retained its elevated evolvability through episodes of diversification and range expansion. Moreover, interestingly, the youngest large radiation in this region, which occurred in Lake Victoria, cannot have started until more than 100 000 years (approx. 50 000 generations) after the original hybridization episode, implying that the hybridization-induced evolvability persisted over a very long period of time [12]. These observations imply that the rare and unlikely event of hybridization between lineages of the right genetic distance may only need to occur once for promoting recurrent ARs in the hybrid lineage.

However, this observation-based hypothesis raises theoretical paradoxes. That is, while hybridization-induced high evolvability needs to persist on the long-term and spread through space for promoting multiple ARs, both are thought to be difficult. First, large genetic variation will be lost in the process of adaptation in response to natural selection because beneficial alleles will become fixed in the emerging species [21]. Hence, species that have originated from a hybrid swarm, after adapting to their specific ecological niches, should no longer retain especially high genetic variation and associated elevated evolvability. Natural selection will also inhibit the geographic spread of hybridization-induced AR. For instance, when a hybrid fish population that originated in a lake expands its range along river corridors and finally reaches a new lake, natural selection by the river environment will filter out a large fraction of genetic variation that was otherwise available for AR in lakes. Second, large standing genetic variation generated by hybridization that exceeds the variation supported at mutation-drift equilibrium will decline over time through genetic drift until the population returns to equilibrium variation [21,22]. Similarly, genetic drift caused by serial founder events during range expansion of a hybrid population is generally expected to prevent transmission of high genotypic diversity to new areas [23]. In sum, a single hybridization event promoting multiple independent ARs seems unlikely because the founder population of secondary radiation, one or a few species from the first radiation, will have lost the elevated adaptive variation due to natural selection and genetic drift before the onset of the second radiation.

Here, we propose hypothetical mechanisms by which exceptional genetic variation once generated by a singular event of hybridization between distant evolutionary lineages can geographically expand and persist for the long term despite the above mechanisms. We consider ecological and geographic mechanisms that promote subdivision of the hybrid population into genetically isolated lineages and subsequent secondary admixture between some of these. It has been known for a long time that population subdivision can prevent global loss by genetic drift of selectively neutral genetic variation [22,24,25], because genes fixing in each sub-lineage by chance will be different between sub-lineages. However, it has also been argued that population subdivision alone is not likely to maintain adaptive (i.e. non-neutral) genetic variation especially when all sub-lineages are subject to similar selection pressures because fixation of the same alleles will be favoured in all sub-lineages [24,25]. Our hypothesis extends these previous population genetic theories by proposing that subdivision of a hybrid population can promote the maintenance of not only neutral but also adaptive genetic variation if ecological traits are polygenic. The reason is that many distinct genotypes can produce similar phenotypes when a trait is polygenic and phenotypic effects of alleles can be compensating one another [26]. Such a redundant genotype–phenotype mapping will allow fixation of different sets of alleles in adaptation-relevant loci even between phenotypically convergent sub-lineages. Subsequent secondary admixture between sub-lineages can restore polymorphisms from the ancestral hybridization and associated high phenotypic variation and evolvability through genetic reshuffling even when each of the hybrid-clade sub-lineages had already lost most phenotypic variation. Moreover, we propose that similar mechanisms can enable not only long-term maintenance but also the geographical spread of hybridization-induced evolvability. For example, if a hybrid population expands its range along multiple geographically isolated dispersal paths, populations expanding along different paths can form genetically distinct sub-lineages. If such sub-lineages meet and hybridize in newly colonized areas, large adaptive genetic variation will instantaneously be reestablished even if each sub-lineage had lost genetic/phenotypic variation and had undergone convergent phenotypic evolution during its range expansion.

The goal of this study is to theoretically clarify geographical and ecological conditions under which genetic variation generated by single hybridization events can persist for long periods and spread to distant areas, facilitating recurrent AR. We develop individual-based models simulating two evolutionary scenarios. The first scenario, 'spatially repeated AR', investigates the recurrent occurrence of ARs in distant regions during a geographical range expansion of a hybrid lineage. The second scenario, 'temporally repeated AR', focuses on the reestablishment of AR after a drastic but geologically transient environmental change had destroyed earlier AR that had evolved from a hybrid population.

## 2. Methods

### (a) Model overview

The model considers diploid organisms with non-overlapping generations. Individuals are either female or male. To keep the model simple so that simulation results will be interpretable, we consider the evolution of only a single quantitative ecological trait $x$. Trait $x$ is controlled by $L$ loci located at random positions in a genome consisting of $2n$ chromosomes. Each chromosome is $l$ base-pairs long. All loci have additive effect; trait value of an individual $i$ is given by $x_i = \sum_{k=1}^{L} (e_{ki1} + e_{ki2})$, where $e_{ki1}$ and $e_{ki2}$ are phenotypic effect values of two alleles of the locus $k$. The phenotypic effect value of alleles can be altered by point mutations that occur at a rate $\mu$ per locus in meiosis. Each allele carries a set $S$ of derived nucleotides and a mutation on the allele adds a new derived nucleotide to $S$. The effect of each derived nucleotide $\varepsilon$ follows the normal distribution $N(0, \sigma_m^2)$. The phenotypic effect value of an allele, $e_{kij}$, is given by $\sum_{u \in S_{kij}} \varepsilon_u$ where $S_{kij}$ is the set of derived nucleotides carried by the allele and $\varepsilon_u$ is the effect of the derived nucleotide $u$. In addition to mutations, crossover recombination, which occurs both between and within loci at a rate $r$ per base-pair in meiosis, creates chromosomes with novel combinations of derived nucleotides. More details of the model are described in the electronic supplementary material, appendix S1.

The model considers multiple habitable patches, each of which contains two or five parapatric habitats (figure 1). There are five types of habitats ($H_{1,2,\ldots,5}$) in which growth performance is maximized with habitat-specific optimal trait values $x_{opt\_H}$ ($x_{opt\_H} = 0$, 5, 10, −5, −10 for $H = H_{1,2,\ldots,5}$). The growth performance $P_i$ of an individual $i$ in a habitat $H$ is given by $\exp\{-\varphi(x_i - x_{opt\_H})^2\}$,

**Figure 1.** Simulation scenarios. Blue squares are habitable patches, each contain five or two habitats. Blue bands connecting patches indicate links between patches by migration of newborn individuals. (*a*) Spatially repeated AR scenario. (*b*) Temporally repeated AR scenario. Details of each scenario are in the Methods. (Online version in colour.)

where $\varphi$ is a parameter controlling the strength of natural selection. Survival probability of an individual $i$ is given by $P_i / \left[ 1 + \left\{ \sum_{j=0}^{M_{Hi}} P_j / (P_i N f / (f-2)) \right\} \right]$, where $M_{Hi}$ is the number of individuals in the same habitat as the individual $i$, $f$ is per-capita female fecundity ($f > 2$), and $N$ is a constant that determines carrying capacity of the habitat. The biological and theoretical basis of this survival function is described in the electronic supplementary material, appendix S1. In short, this survival function assumes that both basal survival rate in the absence of competition and superiority in competition increases linearly with individuals' growth performance. Each surviving female selects a single mating partner randomly from males living in the same habitat of the same patch. Newborn individuals move to a neighbouring patch with probability $m_P$ and to a randomly selected habitat in the same patch with probability $m_H$. We used $m_H = 0.2$, with which gene flow between populations in parapatric habitats is weakly limited. Gene flow among populations in distinct habitats can be further reduced through adaptive differentiation of the ecological trait $x$ because immigrant inviability, the increased mortality of immigrants with locally maladaptive phenotypes [27], can reduce effective migration rate. Therefore, divergent ecological adaptation leads to incipient ecological speciation unless the natural selection is too weak, although the incipient species will lack permanent reproductive isolation and can merge if spatial isolation among habitats or divergent natural selection is removed.

## (b) Simulation of hybridization

Prior to simulating evolutionary dynamics caused by hybridization, we prepared two parental lineages for hybridization by simulating genomic evolution of two isolated populations starting from one common ancestor. In the common ancestral genome, effects of all alleles were set to 0 ($e_{ki1} = e_{ki2} = 0$ for any $k$). Two populations evolved independently for $T_0$ generations under identical habitat conditions ($H_1$) in which the optimal trait value was 0 and the selection strength was $\varphi_P$ (figure 1). Although phenotypes of both lineages remained near $x = 0$ throughout $T_0$ generations, their genomes independently fixed many mutations with compensating phenotypic effects (electronic supplementary material, figure S1). The accumulated genetic differences between lineages can give rise to novel phenotypic variation when genomes of two lineages are recombined in a hybrid population (e.g.

transgressive segregation) [16]. Secondary contact and hybridization of the parental lineages were simulated by introducing 100 individuals from each population into the same patch. Our preliminary simulations confirmed that hybridization facilitates a single episode of AR under a wide range of parameter conditions (electronic supplementary material, appendix S2 and figure S2), including our default parameter set (electronic supplementary material, table S1; empirical basis of default parameter values is described in the electronic supplementary material, appendix S3).

In simulations in the remaining parts of this paper, we limited the source of genetic variation to the initial hybridization event. That is, we did not allow any spontaneous mutations after the period of allopatric evolution and ruled out the effect of standing genetic variation within parental populations by simulating hybridization between two clonal populations each composed of a single haploid genome randomly sampled from one of two parental populations. Owing to these assumptions, if recurrent AR occurs in our simulations, we can safely conclude that genetic variation generated by the initial hybridization event can promote recurrent AR.

## (c) Spatially repeated adaptive radiation scenario

The spatially repeated AR scenario considered two large patches (region 1 and 2) connected by two lines of expansion corridors, each of which consists of $k$ tandemly connected small patches ($CP_{i,j}$; $i = 1, 2$; $j = 1, 2, \ldots, k$) (figure 1*a*). This landscape structure represents, for example, two large lakes connected by rivers. Each of regions 1 and 2 contained five distinct habitats ($H_{1,2,\ldots,5}$), whereas each corridor patch contained two habitats $H_A$ and $H_B$. We assumed $H_A = H_B = H_1$ for cases with an environmentally homogeneous corridor, whereas $H_A = H_1$ and $H_B = H_3$ for cases with an environmentally heterogeneous corridor. In cases with a single corridor, $CP_{1,j}$ and $CP_{2,j}$ were connected by migration at rate 0.5 for any $j$ (figure 1*a*, yellow arrows), whereas in cases with two geographically isolated corridors, there was no migration between $CP_{1,j}$ and $CP_{2,j}$. Natural selection strength $\varphi$ was set to $\varphi_B$ in regions 1, 2 and $\varphi_C$ in corridor patches. Each simulation was started by introducing two parental lineages into habitat 1 of region 1 and continued for 5000 generations after the colonization.

We explored geographic and ecological conditions under which genetic variation generated by hybridization occurring in the region 1 can promote recurrent AR in both region 1 and 2. For this purpose, we performed simulations systematically varying values of the following five parameters: the number of geographically isolated expansion corridors (1 or 2), the number of distinct habitats in each corridor (1 or 2), the length of the corridor(s) ($k$), the strength of stabilizing selection in corridor patches ($\varphi_C$) and the carrying capacity of corridor patches ($N_C$). With each parameter set, we performed 30 simulation replications. For this analysis, we first simulated evolution of 30 independent pairs of two parental lineages with the default parameter set (electronic supplementary material, table S1). Then, for all parameter sets, we performed 30 simulations of hybridization and evolution that follow using the same set of 30 parental lineage pairs to form the initial hybrid populations.

## (d) Temporally repeated adaptive radiation scenario

The temporally repeated AR scenario considered three patches, each of which contains five types of habitat ($H_{1, 2, \ldots 5}$). Three patches form a single large region as they are connected by the migration of individuals at a high rate ($m_p = 0.1$). We simulated a geologically transient environmental change, such as climate change or geographic dynamics, that causes contraction and subdivision of the habitable zone and loss of divergent ecological selection (figure 1b). We modelled transient environmental change as a reduction of values of the following parameters: (i) the maximum carrying capacity in single habitat (from $N_B$ to $N_R$), (ii) migration rate between three patches (from $m_P$ to $m_R$) and (iii) strength of natural selection (from $\varphi_B$ to $\varphi_R$). We used $\varphi_R = 0$ (i.e. no selection), and thus the environmental change always led to the loss of ecological diversity that has evolved before the environmental change. Changes in values of these three parameters proceeded gradually between generations 10 000 and 10 100. The altered environmental condition continued for the following 9900 generations ('the refugial phase'). Then, a renewed environmental change took place between generations 20 000 and 20 100 during which the values of the three parameters returned to the original values ($N_B$, $m_B$, $\varphi_B$). Each simulation was continued until generation 30 000.

To investigate whether and under what conditions AR can be reestablished after a collapse by the environmental change, we conducted simulations systematically varying values of the basal strength of natural selection $\varphi_B$ and the migration rate between three patches and carrying capacity of single habitats in the refugial phase $m_R$ and $N_R$.

# 3. Results

## (a) The spatially repeated adaptive radiation scenario

In all our simulation runs, hybridization between two colonizing lineages in region 1 was followed by rapid AR within region 1. However, when there was only a single corridor without environmental heterogeneity, AR was not repeated in region 2 unless two regions were close to each other (small $k$) (figures 2c and 3a; electronic supplementary material, S3a). This was because the high genetic/phenotypic variation generated by hybridization was not maintained in the corridor population due to the combination of stabilizing selection in corridor patches and genetic drift associated with successive founder events at the front of the range expansion. Consequently, the lack of genetic/phenotypic variation prevented AR in the second region despite ecological opportunity afforded by the available resources and environmental heterogeneity. In line with this interpretation, the likelihood

of recurrent AR was further suppressed both by increasing the strength of stabilizing selection in corridor patches and by decreasing the population size in corridor patches (figure 3a; electronic supplementary material, figure S4a).

By contrast, AR did occur in region 2 even after a long-distance range expansion when there were two geographically isolated expansion corridors (figures 2a and 3b; electronic supplementary material, S3b). Isolation of two range expansion waves in these corridors led to the formation of two genetically distinct sub-lineages because the populations in the two corridors fixed different sets of alleles during their range expansion. Although phenotypic/genetic variation eroded within both sub-lineages during range expansion, secondary admixture between them in region 2 reestablished high standing genotypic/phenotypic variation, which permitted AR in region 2.

AR in region 2, however, often failed to occur when one expansion front reached region 2 well before the other expansion front. In this situation, migration from region 2 into the second corridor led to the situation where the contact and hybrid zone between the two lineages resides inside one of the corridors rather than in region 2 (electronic supplementary material, figure S5). Despite the presence of ecological opportunity in region 2, no radiation occurs because the necessary genetic variation never arrives there. Similarly, our additional simulations revealed that the recurrence of AR mediated by two isolated expansion corridors was hindered if the parental lineage 1 was introduced to region 1 and spread across the entire system of connected patches before the parental lineage 2 arrived in region 1 (electronic supplementary material, figures S6b and S7). In this case, the arrival of lineage 2 in region 1 led to hybridization-induced AR there, but genetic material from the parental lineage 2 did not spread through corridors to region 2. This was because most of the hybrid genotypes and associated phenotypes were maladaptive in corridor patches compared to those of the resident lineage 1 population. However, our additional simulations revealed that admixture variation can spread to region 2 if migration in corridors is unidirectional (electronic supplementary material, figure S8). Unidirectionality of migration is often the case in dendritic river–lake systems [28].

Environmental heterogeneity within a single dispersal corridor could also facilitate recurrent AR even with long corridors (figures 2b and 3c; electronic supplementary material, figure S3c). Corridors with two environmentally distinct habitats allowed two ecologically distinct incipient species, which had arisen in the AR of hybrid origin in region 1, to independently disperse to region 2 through the different habitats in the corridor. The two incipient species could, nonetheless, form a hybrid population upon the invasion of unoccupied habitats in region 2 because immigrant inviability, the only mechanism for ecological speciation in our model, is strongly context-dependent and is absent in habitats that hold no locally adapted resident competitors. Admixture in these cases reestablished phenotypic variation that often exceeded the range of that of the two incipient species combined (i.e. transgressive segregation) and permitted the evolution of new species adapting to habitats $H_2$, $H_4$ and $H_5$. Different from the case with two geographically isolated corridors, recurrent AR mediated by an environmentally heterogeneous corridor was not hindered by the time lag between the introductions of two parental lineages (electronic supplementary material, figure S6c) because habitat $H_3$ in the corridor had not been occupied until a specialist

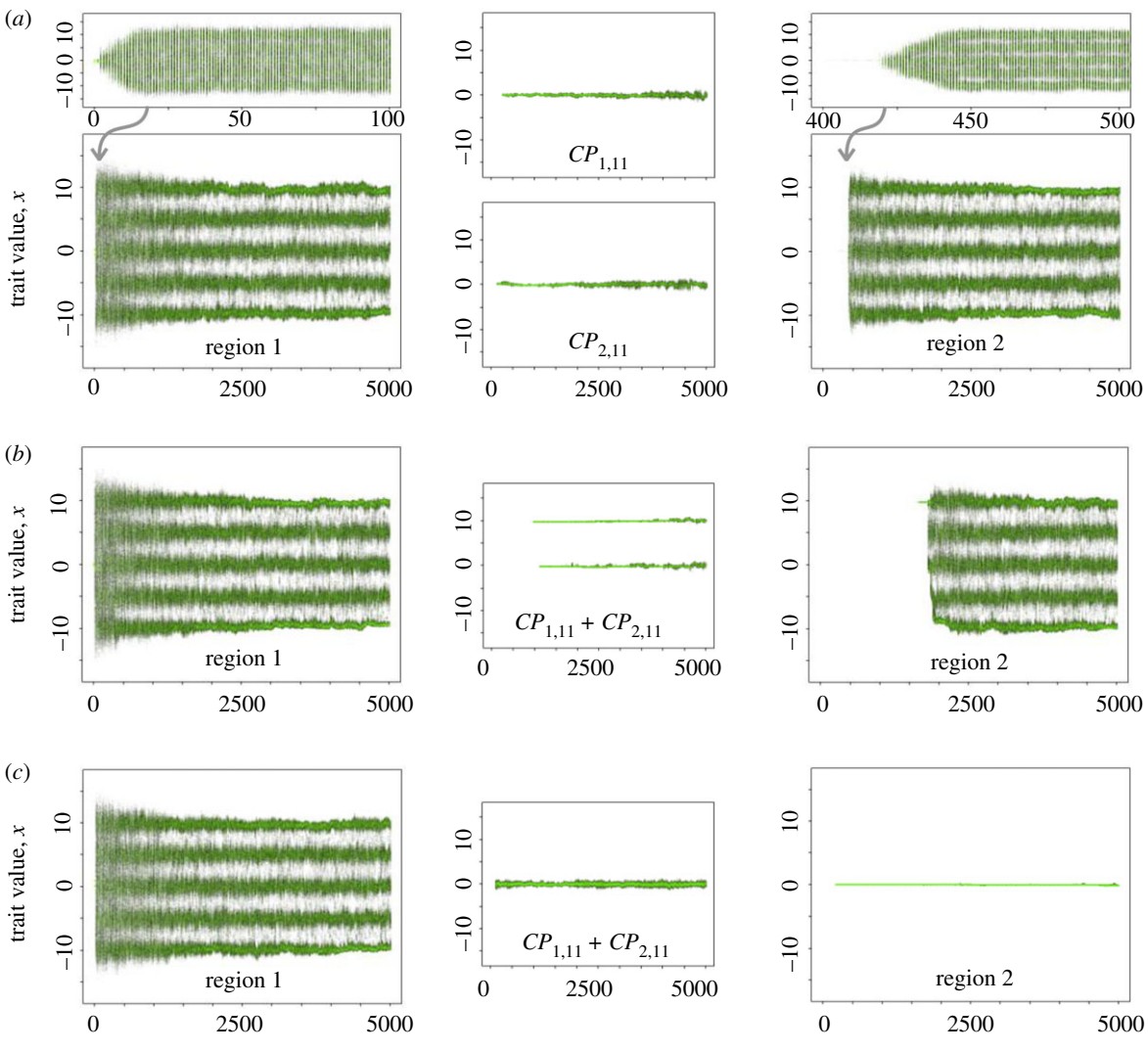

**Figure 2.** Simulation examples with the spatially repeated AR scenario. (*a*) With two geographically isolated corridors without environmental heterogeneity. Upper plots show magnifications of evolutionary dynamics immediately after colonization of region 1 and 2. (*b*) With single corridor with environmental heterogeneity. (*c*) With single corridor without environmental heterogeneity. Trait *x* of all individuals in region 1, 2 and the middle patch of corridor(s) are shown for each generation. Individuals who did and did not survive are shown in green and grey respectively. Corridor length: $k = 20$; other parameters: default values in electronic supplementary material, table S1.

of this habitat evolved through the hybridization-enabled AR in region 1.

For the above simulations, we assumed that regions 1 and 2 had the same set of habitats (or niches). With supplementary simulations, we confirmed that our conclusions are valid also when (i) regions 1 and 2 have different sets of habitats with different fitness optima (electronic supplementary material, figure S9), and when (ii) the ecological trait axis subjected to divergent selection in the region 2 is different from that in the region 1 (electronic supplementary material, figure S10).

## (b) The temporally repeated adaptive radiation scenario
Our simulations demonstrated that ecological speciation and geographical isolation of lineages can both facilitate long-term maintenance of hybridization-induced adaptive genetic variation, thereby enabling a renewed hybridization-enabled AR after the transient environmental change had destroyed earlier AR. Figure 4*a* shows an example of evolutionary dynamics. Hybridization of two parental lineages produced large standing genotypic/phenotypic variation leading to rapid AR. This AR, however, collapsed after environmental

changes caused the relaxation of divergent ecological selection and the associated loss of reproductive isolation. Formerly ecologically differentiated incipient species merged into a hybrid swarm and lost their ecological differentiation (i.e. speciation reversal). As three patches (hereafter, refugia) were isolated during the refugial phase, different sets of genes became fixed in the three refugial patches. Although standing genetic variation within each refugial sub-lineage was lost during the refugial phase through genetic drift (figure 4*a*, generations 10 000–20 000), subsequent admixture between them after reconnecting of the refugial patches recovered large standing genetic/phenotypic variation, which was available to natural selection when environmental conditions returned to the original state (generations 20 000–30 000 approx.). Consequently, AR could be reestablished quickly despite the absence of spontaneous mutations.

Simulations with varying parameter values revealed conditions under which AR can be reestablished after the collapse of original radiation. To evaluate the sustainability of high evolvability under the conditions given by different parameter combinations, we tracked the number of polymorphic nucleotide sites (SNPs) among all individuals in

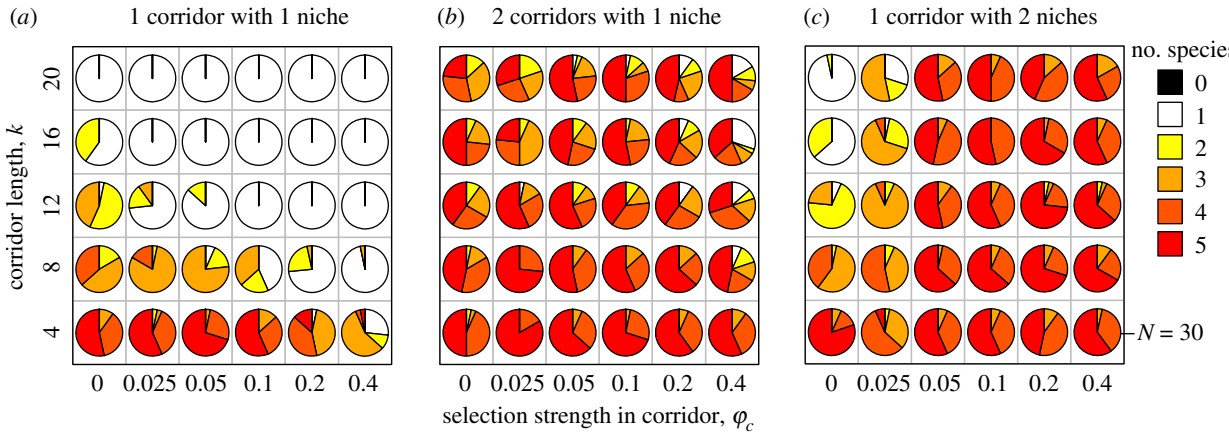

**Figure 3.** Conditions for the spatially repeated AR. (*a*) With single corridor without environmental heterogeneity. (*b*) With two geographically isolated corridors without environmental heterogeneity. (*c*) With single corridor with environmental heterogeneity. Each pie chart shows frequencies of the number of incipient species in region 2 after 5000 generations from the initial hybridization event. Other metrics of evolutionary diversification gave consistent results (electronic supplementary material, figure S3). Other parameters: default values in electronic supplementary material, table S1. (Online version in colour.)

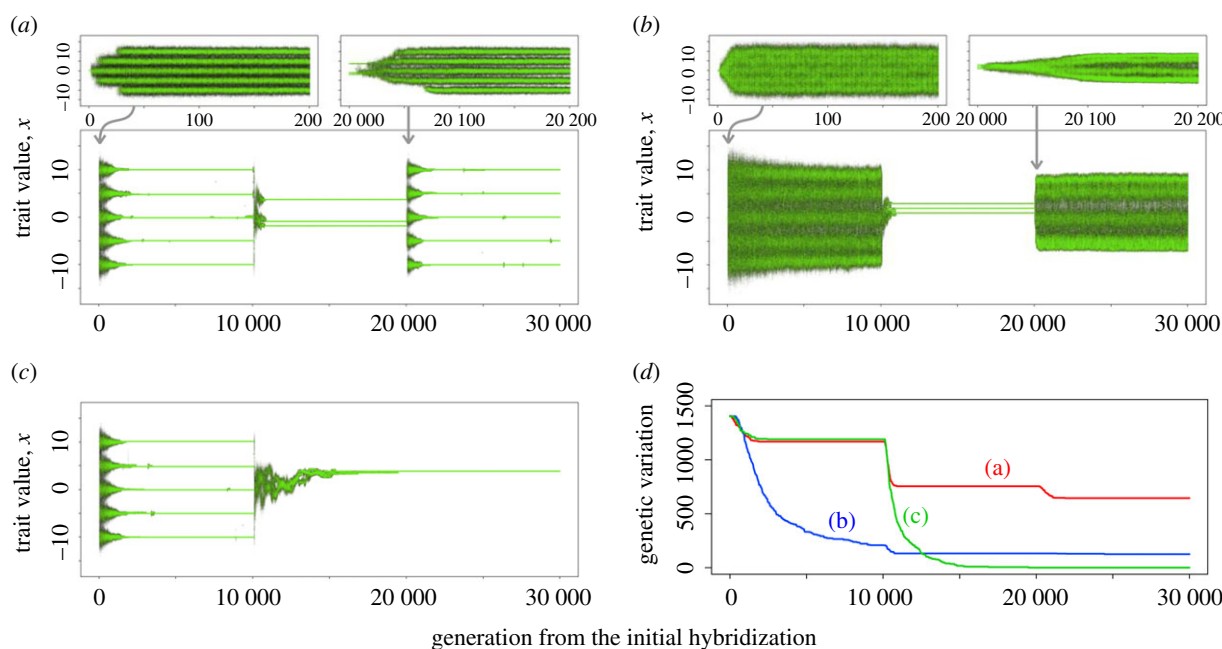

**Figure 4.** Simulation examples with the temporally repeated AR scenario. (*a*) With strong divergent selection in AR ($\varphi_B = 0.2$) and complete isolation of refugia ($m_R = 0$). (*b*) With weak divergent selection in AR ($\varphi_B = 0.0125$) and complete isolation of refugia ($m_R = 0$). (*c*) With strong divergent selection in AR ($\varphi_B = 0.2$) and incomplete isolation of refugia ($m_R = 0.001$). (*d*) Numbers of remaining SNPs at each generation in the three simulation examples. Other parameters: default values in electronic supplementary material, table S1.

the entire meta-community. Hybridization-induced genetic variation could be lost at the meta-community level in two different phases. First, in the period before the refugial phase (generations 0–10 000) if divergent ecological selection between habitats was too weak to generate reproductive isolation between ecologically differentiating sub-lineages (small $\varphi_B$; figures 4*b,d* and 5*b*). By contrast, a large fraction of SNPs could persist throughout this period if strong divergent selection between habitats generated reproductive isolation between ecologically differentiating sub-lineages (large $\varphi_B$; figures 4*a,d* and 5*b*). Second, genetic variation at the meta-community level could decline during the refugial phase (generations 10 000–20 000) due to global fixation of same genes by genetic drift when there was gene flow between the patches (large $m_R$; figures 4*c,d* and 5*b*). This effect could be slightly mitigated if refugia had large population sizes

(large $N_R$; electronic supplementary material, figure S11*c*). If the three patches were strongly isolated (small $m_R$), many polymorphisms could persist in the form of genetic differentiation between sub-lineages (figures 4*d* and 5*b*). Under the conditions where ecological and geographical isolation of lineages could maintain large fractions of the old admixture variation, AR was reestablished when the environment returned to the original state (generations 20 000–30 000 approx.) (figure 5*a*; electronic supplementary material, figures S11 and S12*a,b*).

Our simulations above assumed the scenario where the loss of divergent ecological selection in the refugial phase is due to relaxation of natural selection in all habitats (and hence loss of divergent selection between habitats). An alternative mechanism for the loss of divergent selection is an environmental change that causes a shift of the optimal

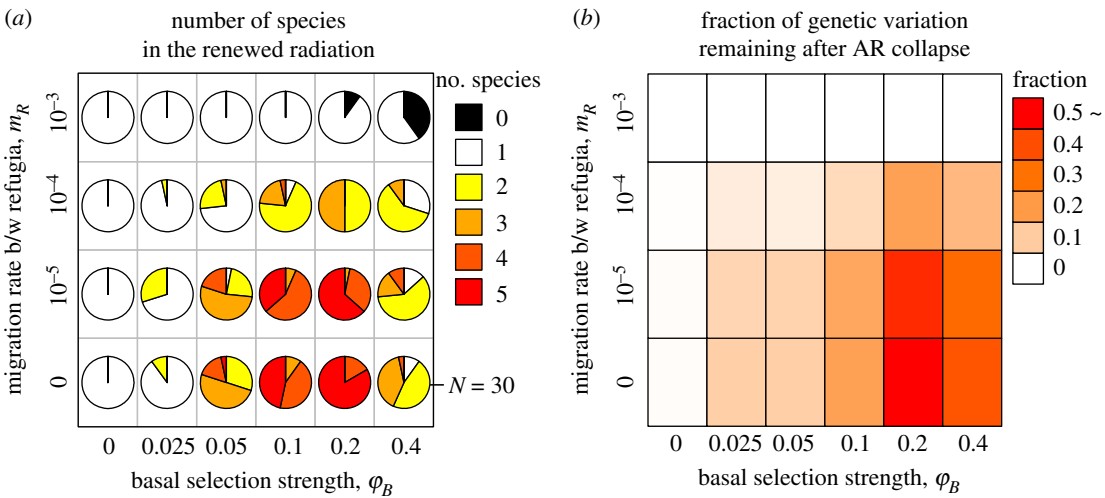

**Figure 5.** Conditions for the temporally repeated AR. (*a*) The number of species at generation 30 000. (*b*) The extent of evolutionary potential at the end of the refugial phase. Each square shows the ratio of the SNPs at generation 20 000 divided by the SNPs at generation 0 (averaged among 30 runs). Other parameters: default values in electronic supplementary material, table S1. (Online version in colour.)

trait values in the different habitats towards the same value. We confirmed that our results were not qualitatively changed under this alternative scenario (electronic supplementary material, figures S13 and S14).

While the number of polymorphisms explained the likelihood of the recurrence of AR well in our simulations, this ignores effects of relics of physical genetic linkage that may have built up adaptively in the first AR. We investigated this in additional simulations that we summarize in electronic supplementary material, figure S11*e*–*h*. These simulations revealed that physical genetic linkage that had built up adaptively may promote recurrent evolution of phenotypes that had once evolved in the past, although the effect was slight in our simulations.

## 4. Discussion

Our results demonstrate that genetic variation generated by a single geographically localized hybridization episode can promote multiple episodes of AR far apart in space and time. Our simulations did not allow any spontaneous mutations after the hybrid swarm origin of the radiating lineage. This way we made sure that the origin of genetic variation that enabled the second AR was the initial hybridization event. If spontaneous mutations are allowed after the initial hybridization event, admixture between sub-lineages will not only restore old but also generate a new genetic variation by reshuffling de novo mutations after they had arisen in isolation in either sub-lineage. In this respect, our simulations are conservative regarding the effects of hybridization between sub-lineages in promoting recurrent radiation. This also means that our results do not generally deny the possibility of recurrent AR with spontaneous mutations alone, although AR was extremely unlikely without hybridization-induced genetic variation under our simulation conditions (electronic supplementary material, figure S2).

The present study extends and integrates the hybrid swarm origin hypothesis of AR and the syngameon hypothesis [20]. The syngameon hypothesis proposes that occasional or locally confined gene flow between species within AR can contribute to maintaining the evolutionary momentum during the AR by replenishing standing genetic variation. Our simulations suggest that rapid formation of a syngameon from a hybrid swarm can enable propagation in space and time of exceptionally high evolvability induced by a single ancestral hybridization event, thereby promoting the repetition of hybrid swarm origin AR.

Another relevant theory is the transporter hypothesis that has been invoked to explain the repeated evolution of freshwater stickleback from marine ancestors. The transporter hypothesis argues that recurrent hybridization between marine and freshwater stickleback may enable transporting of freshwater alleles among geographically isolated freshwater habitats through the spatially contiguous marine population allowing the latter to recurrently invade freshwater habitats [29] through re-evolution of freshwater phenotypes by reuse of the same alleles that had been positively selected in the donor freshwater species or population. This mechanism will operate most efficiently when ecological traits are controlled by single or a few major genes. Whereas the transporter hypothesis can explain how occasional hybridization facilitates parallel evolution, our theory wants to explain how reticulate evolutionary dynamics facilitate the repeated occurrence of AR which may involve some parallel evolution but also the evolution of much novelty. In our model, a secondary admixture of sub-lineages in a place distant from the original hybridization episode generates a wide range of phenotypes, including some with low fitness in the first radiation (electronic supplementary material, figures S9 and S10). This led to the re-establishment of increased evolvability, rather than the re-evolution of a formerly positively selected phenotype. A key for this mechanism is the polygenic control of ecological traits with which the same phenotype can be generated by many different genotypes.

### (a) Empirical implications

Simulations of the spatially repeated AR scenario provide a theoretically feasible explanation for the highly recurrent occurrence of AR in one lineage of African cichlid fish in the Lake Victoria region [12] and also in the Alpine whitefish [6]. In both cases, single hybrid lineages have radiated into monophyletic flocks of species in several different lakes,

while other related and/or ecologically similar lineages of fish in the same region never radiated. Reciprocal monophyly of ecologically diverse species assemblages in different lakes implies that only one species from the first radiation expanded its range to colonize additional lakes through rivers connecting the lakes and subsequently underwent local ARs within each lake. However, this scenario also implies that populations that colonized new lakes could have imported into new lakes only a limited fraction of ecologically relevant genetic variation from the original hybrid population. Yet this seems to conflict with the consistently large genetic variation [30] and the exceptionally high evolutionary potential to form many new species [12]. Our model may resolve this paradox: we have theoretically demonstrated that hybridization-enabled ARs can be recurrent in space and time if either (i) there are multiple range expansion corridors connecting the sites of radiations or (ii) there are more than one habitat niches for the expanding lineage within the dispersal corridor. Indeed multiple range expansion or dispersal corridors have connected several of the great lakes in the Lake Victoria region [12] (O.S. 2020, unpublished data). The presence of either of the two mechanisms can be tested by palaeo-geographic/palaeoclimatic or genetic reconstruction of dispersal paths between radiation sites and by estimating palaeo-ecological communities in dispersal path habitats either by studying fossils or modern analogues.

The spatially repeated AR mediated by colonization through two independent expansion corridors failed when range expansion in one of the parental lineages occurred already before its localized hybridization with the second lineage (electronic supplementary material, figure S6b and S7). This result suggests that spatially repeated AR is more likely to occur when hybridization itself triggers range expansion [31,32] or when hybridization occurs during a range expansion or range shift of both parental lineages [33,34].

Simulations of the temporally repeated AR scenario offer a mechanistic hypothesis for the understanding of some ARs of African lake cichlid fish in which mass extinction due to lake desiccation and re-evolution of species diversity are thought to have occurred. All African Great Lakes experienced massive water level fluctuations driven by climate oscillations [35]. The associated oscillations in the relationship between precipitation and evaporation caused dramatic changes to the ecosystems. The latter ranged from transitions between open-lake freshwater and closed-lake saline conditions with drastic community change and perhaps mass extinctions, as in Lake Malawi [36], to repeated episodes of total desiccation, as in the case of Lake Victoria [37]. Yet even the youngest contemporary freshwater lakes host largely monophyletic radiations of the same haplochromine lineage [38], and the younger of these radiations tend to be phylogenetically nested in older radiations [2,12]. This implies the recent renewal of AR from refugial survivors and/or immigrants from older radiations. Genomic evidence suggests that the AR in modern Lake Victoria was fuelled by genetic variation that was generated during an ancient hybridization event [12] which occurred at least 100 000 years earlier than the onset of this radiation. Our simulation results suggest that admixture between species that by themselves had evolved in the distant past from a hybrid population is indeed a possible mechanism underlying the renewed release of much of the original hybrid swarm variation which would then have facilitated the extremely recent and rapid AR in Lake Victoria and several smaller lakes in the region. In addition to several fish radiations in freshwater lakes, ancestral hybridization events could potentially have facilitated the repeated occurrence of AR of plants and animals in some archipelagos and high mountains where geographic dynamics and climate changes had caused temporal isolation and subsequent secondary admixture of lineages (reviewed in the electronic supplementary material, appendix S5).

## 5. Conclusion

While the evidence is growing that hybridization can be a source of genetic variation that has facilitated adaptive radiations [5,10,11–15,17–19,34] and has sometimes resulted in hybrid lineages that seeded more than one radiation [6,12], it remained obscure how hybridization could facilitate recurrent ARs long after the hybridization had taken place and in distant places. Our simulation study suggests that speciation and geographic isolation and subsequent secondary admixture of species or sub-lineages within a hybrid clade can enable geographic expansion and long-term maintenance of hybridization-induced genetic variation which exceeds mutation–selection–drift balance. These results suggest that genetic variation generated by single episodes of hybridization, spatially narrowly confined, can affect macro-voluntary processes at large spatial and temporal scales, such as phylogenetically nested recurrent ARs.

Data accessibility. Program source codes for computer simulations and simulation results for each figure are deposited on the Dryad Digital Repository: https://doi.org/10.5061/dryad.nzs7h44pv [39].

Authors' contributions. K.K. and O.S. designed the study; K.K. developed the computer simulation model and conducted analyses; K.K. wrote the manuscript together with O.S.

Competing interests. We declare we have no competing interests.

Funding. This work was supported by the JSPS Overseas Research Fellowship and the JSPS Research Fellowships for Young Scientists for K.K.

Acknowledgement. We thank Joana Meier for discussions during the development of ideas for this study, and Philine Feulner, David Frei, Cas Retel, David Marques, Ayana Martins, Chad Brock, Oliver Selz, Carlos Melian, Blake Matthews, Julian Junker and members of the Fish Ecology & Evolution department of EAWAG for fruitful discussions.

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
