## [Reviewer comments · Proceedings of the Royal Society B: Biological Sciences]

Review History

RSPB-2020-0941.R0 (Original submission)

Review form: Reviewer 1

Recommendation

Major revision is needed (please make suggestions in comments)

Scientific importance: Is the manuscript an original and important contribution to its field?

Good

General interest: Is the paper of sufficient general interest?

Good

Quality of the paper: Is the overall quality of the paper suitable?

Good

Is the length of the paper justified?

Yes

Should the paper be seen by a specialist statistical reviewer?

No

Do you have any concerns about statistical analyses in this paper? If so, please specify them explicitly in your report.

No

It is a condition of publication that authors make their supporting data, code and materials available - either as supplementary material or hosted in an external repository. Please rate, if applicable, the supporting data on the following criteria.

Is it accessible?

Yes

Is it clear?

Yes

Is it adequate?

Yes

Do you have any ethical concerns with this paper?

No

Comments to the Author

Kagawa and Seehausen use individual-based simulations to explore scenarios under which recurrent adaptive radiations can occur following an initial hybridisation event. They describe several mechanisms that can maintain variation in a polygenic trait, which can later be recombined and lead to further radiation. Adaptive radiations are valuable for understanding the origins of diversity, but their empirical complexity can be overwhelming. Simulation studies like this one that reduce the problem to generalisable mechanisms can be very useful in aiding our understanding.

I think the simulation design is thorough and well reasoned, and the findings are generally well presented. I have one general concern about the framing of the novelty of the paper, and several specific concerns that are all minor, although one does suggest an additional set of simulations.

General concerns

To me the novelty of this study is fairly subtle, and is not sufficiently clear from the abstract. There is much recent work (cited in this study) showing that combining standing adaptive variation can facilitate speciation and adaptive radiation. The process underlying recurrent radiations in the present paper is, in my opinion, not a distinct process. Just like the first radiation, the second radiation is triggered by bringing together old adaptive variation. I realise that the variation has already been through an initial hybridisation event, but I don't think that fundamentally changes the nature of the second radiation. Therefore, the main findings of this study are the mechanisms by which sufficient adaptive variation can be maintained: essentially population structure either by physical isolation or heterogeneous habitat. I worry that using the term "dynamic fission and fusion", rather than the familiar terms above, might artificially inflate the distinctiveness of the mechanisms presented.

To be clear, I do still think this is a valuable study. It has certainly helped me better understand the circumstances under which pre-existing variation may spawn multiple radiations. There are also some nice nuanced points that could receive more emphasis: (1) The fact that the trait is polygenic and there is a distribution of fitness effects means that even isolated lineages under the same selective regime will maintain distinct variation. (2) A second radiation can fail to occur in a new habitat if the colonising lineages expand at different rates.

In summary, I think the paper (especially the abstract) would be improved if the authors were more specific and up front about the fact that this paper is not presenting an entirely new

evolutionary process, it is exploring circumstances under which the pre-existing variation can be maintained and recombined, such that the same process can be repeated.

Specific comments

1. Lines 319-329 argue that the process seen here is distinct from the transporter hypothesis, because “secondary admixture of sub-lineages in a place distant from the original hybridisation episode generated a wide range of phenotypes, including those with low fitness in the first radiation”. I agree that the process here is distinct from the transporter hypothesis, because the genotypic clusters that emerged in the second radiation tended to be distinct from those in the first radiation, but I do not see what part of the results shows that the second radiation involved phenotypes that would have low fitness in the first radiation. Surely to test that idea one would require the second radiation to have a distinct set of optima? I think including such a model would be a nice addition to the paper.
2. The very first sentence of the Results (line 186) is quite long and difficult to follow, and I also found it a bit misleading. It is not clear on what is meant by long-distance, nor does it account for the fact that expansion along a single corridor can lead to a second AR, provided the corridor has a heterogeneous environment. My first thought was that the entire sentence is unnecessary, but if you would like to start with a summary, I suggest a more general statement that the occurrence of a second AR depended on the maintenance of variation during expansion, which could occur in a few different ways.
3. Line 99: “crossover recombination, which occurs at a rate r per base-pair in meiosis, creates chromosomes with novel combinations of derived nucleotides”. Although more detail is provided in the Appendix, I think it is necessary to mention here that recombination can occur within loci (i.e. to separate derived mutations at the same locus), as this is not obvious from the paragraph as it is currently written.
4. Appendix S1: “Every derived nucleotide has a unique randomly assigned position within its target locus”. Does this mean it is an infinite sites mutation model, such that recurrent mutation of a single site is not possible? Is this reasonable given the time scales and mutation rates being considered?
5. Figure S1 I think it might be clearer to replace “mutations accumulated” with “mutations fixed” or simply “substitutions”.
6. Figure S2 legend explains why an initial radiation does not produce many species when selection is weak, but it does not explain why very strong selection also fails to produce an initial radiation following hybridisation. Is this because some habitats are never colonised?
7. Figure 2 (and others like it) is too small when drawn across a single pdf page.
8. Effect of breaking linkage structures (lines 271-269). To me, comparing Figures 5 and S12, it is very difficult to make out whether or not there is a difference. Could this be described quantitatively?
9. Lines 289-293 argue that, in addition to standing adaptive variation, new mutations that arise in the separated lineages would further facilitate future AR. I partly agree, but I think one exception is cases where the temporary isolation involves complete breakdown of selection. The key aspect of the older standing variation is that it arose in a heterogeneous environment with selection. Mutations that fix in the absence of selection will surely not facilitate future AR, otherwise we would expect any increase in variation to facilitate future AR, such as increasing the population size, right?

10. Line 347: it would be helpful to add some information on whether there is evidence that either multiple corridors or heterogeneous corridors existed in these empirical cases.

11. Line 367: This should refer to Appendix S5.

Review form: Reviewer 2 (Robert Gilman)

Recommendation

Accept with minor revision (please list in comments)

Scientific importance: Is the manuscript an original and important contribution to its field?

Excellent

General interest: Is the paper of sufficient general interest?

Good

Quality of the paper: Is the overall quality of the paper suitable?

Good

Is the length of the paper justified?

Yes

Should the paper be seen by a specialist statistical reviewer?

No

Do you have any concerns about statistical analyses in this paper? If so, please specify them explicitly in your report.

No

It is a condition of publication that authors make their supporting data, code and materials available - either as supplementary material or hosted in an external repository. Please rate, if applicable, the supporting data on the following criteria.

Is it accessible?

N/A

Is it clear?

N/A

Is it adequate?

N/A

Do you have any ethical concerns with this paper?

No

Comments to the Author

See attachment. (See Appendix A)

Decision letter (RSPB-2020-0941.R0)

01-Jul-2020

Dear Dr Kagawa:

Your manuscript has now been peer reviewed and the reviews have been assessed by an Associate Editor. The reviewers' comments (not including confidential comments to the Editor) and the comments from the Associate Editor are included at the end of this email for your reference. As you will see, the reviewers and the Editors have raised some concerns with your manuscript and we would like to invite you to revise your manuscript to address them.

Research ethics:

Use of animals and field studies:

Please submit a copy of your revised paper within three weeks. If we do not hear from you within this time your manuscript will be rejected. If you are unable to meet this deadline please let us know as soon as possible, as we may be able to grant a short extension.

Best wishes,
Dr Daniel Costa
mailto: proceedingsb@royalsociety.org

Associate Editor

Board Member: 1

Comments to Author:

Both Referees were supportive of a simulation study that explored scenarios under which recurrent adaptive radiations could occur following hybridisation event(s) and investigating the mechanisms that could maintain variation in a polygenic trait and contribute to further radiations. The Referees provide very detailed reviews that would need to be addressed, including additional simulations (Referee 1), and I agree with the Referees that the novelty of the study was compromised as presented. Nonetheless, the authors present a proof-of-concept model of this hypothesis that could be clear, convincing and an excellent fit in Proceedings B (Referee 2). A thorough revision addressing these and other points raised is needed to evaluate and address the concerns raised.

Reviewer(s)' Comments to Author:

Referee: 1

Comments to the Author(s)

Kagawa and Seehausen use individual-based simulations to explore scenarios under which recurrent adaptive radiations can occur following an initial hybridisation event. They describe several mechanisms that can maintain variation in a polygenic trait, which can later be recombined and lead to further radiation. Adaptive radiations are valuable for understanding the origins of diversity, but their empirical complexity can be overwhelming. Simulation studies like this one that reduce the problem to generalisable mechanisms can be very useful in aiding our understanding.

I think the simulation design is thorough and well reasoned, and the findings are generally well presented. I have one general concern about the framing of the novelty of the paper, and several specific concerns that are all minor, although one does suggest an additional set of simulations.

General concerns

To me the novelty of this study is fairly subtle, and is not sufficiently clear from the abstract. There is much recent work (cited in this study) showing that combining standing adaptive variation can facilitate speciation and adaptive radiation. The process underlying recurrent radiations in the present paper is, in my opinion, not a distinct process. Just like the first radiation, the second radiation is triggered by bringing together old adaptive variation. I realise that the variation has already been through an initial hybridisation event, but I don't think that fundamentally changes the nature of the second radiation. Therefore, the main findings of this study are the mechanisms by which sufficient adaptive variation can be maintained: essentially population structure either by physical isolation or heterogeneous habitat. I worry that using the term "dynamic fission and fusion", rather than the familiar terms above, might artificially inflate the distinctiveness of the mechanisms presented.

To be clear, I do still think this is a valuable study. It has certainly helped me better understand the circumstances under which pre-existing variation may spawn multiple radiations. There are also some nice nuanced points that could receive more emphasis: (1) The fact that the trait is polygenic and there is a distribution of fitness effects means that even isolated lineages under the same selective regime will maintain distinct variation. (2) A second radiation can fail to occur in a new habitat if the colonising lineages expand at different rates.

In summary, I think the paper (especially the abstract) would be improved if the authors were more specific and up front about the fact that this paper is not presenting an entirely new evolutionary process, it is exploring circumstances under which the pre-existing variation can be maintained and recombined, such that the same process can be repeated.

Specific comments

1. Lines 319-329 argue that the process seen here is distinct from the transporter hypothesis, because "secondary admixture of sub-lineages in a place distant from the original hybridisation episode generated a wide range of phenotypes, including those with low fitness in the first radiation". I agree that the process here is distinct from the transporter hypothesis, because the genotypic clusters that emerged in the second radiation tended to be distinct from those in the first radiation, but I do not see what part of the results shows that the second radiation involved phenotypes that would have low fitness in the first radiation. Surely to test that idea one would require the second radiation to have a distinct set of optima? I think including such a model would be a nice addition to the paper.

2. The very first sentence of the Results (line 186) is quite long and difficult to follow, and I also found it a bit misleading. It is not clear on what is meant by long-distance, nor does it account for the fact that expansion along a single corridor can lead to a second AR, provided the corridor has

a heterogeneous environment. My first thought was that the entire sentence is unnecessary, but if you would like to start with a summary, I suggest a more general statement that the occurrence of a second AR depended on the maintenance of variation during expansion, which could occur in a few different ways.

3. Line 99: "crossover recombination, which occurs at a rate r per base-pair in meiosis, creates chromosomes with novel combinations of derived nucleotides". Although more detail is provided in the Appendix, I think it is necessary to mention here that recombination can occur within loci (i.e. to separate derived mutations at the same locus), as this is not obvious from the paragraph as it is currently written.

4. Appendix S1: "Every derived nucleotide has a unique randomly assigned position within its target locus". Does this mean it is an infinite sites mutation model, such that recurrent mutation of a single site is not possible? Is this reasonable given the time scales and mutation rates being considered?

5. Figure S1 I think it might be clearer to replace "mutations accumulated" with "mutations fixed" or simply "substitutions".

6. Figure S2 legend explains why an initial radiation does not produce many species when selection is weak, but it does not explain why very strong selection also fails to produce an initial radiation following hybridisation. Is this because some habitats are never colonised?

7. Figure 2 (and others like it) is too small when drawn across a single pdf page.

8. Effect of breaking linkage structures (lines 271-269). To me, comparing Figures 5 and S12, it is very difficult to make out whether or not there is a difference. Could this be described quantitatively?

9. Lines 289-293 argue that, in addition to standing adaptive variation, new mutations that arise in the separated lineages would further facilitate future AR. I partly agree, but I think one exception is cases where the temporary isolation involves complete breakdown of selection. The key aspect of the older standing variation is that it arose in a heterogeneous environment with selection. Mutations that fix in the absence of selection will surely not facilitate future AR, otherwise we would expect any increase in variation to facilitate future AR, such as increasing the population size, right?

10. Line 347: it would be helpful to add some information on whether there is evidence that either multiple corridors or heterogeneous corridors existed in these empirical cases.

11. Line 367: This should refer to Appendix S5.

Referee: 2

Comments to the Author(s)
See attachment

Author's Response to Decision Letter for (RSPB-2020-0941.R0)

See Appendix B.

Decision letter (RSPB-2020-0941.R1)

19-Aug-2020

Dear Dr Kagawa

I am pleased to inform you that your manuscript entitled "The propagation of admixture-derived adaptive radiation potential" has been accepted for publication in Proceedings B.

Open Access

Paper charges

Sincerely,

Dr Daniel Costa

Associate Editor:

Board Member

Comments to Author:

I have now reviewed the revised MS and responses to the previous reviews. The authors have made careful and impactful changes to the paper in line with these reviews and carried out additional simulations with those results strengthening the support for their conclusions. The

result of these revisions is a novel and robust simulation study on mechanisms for the maintenance and spatial spread of ancient admixture variation that will be a very nice contribution to studies of adaptive divergence/radiations. This paper will be of broad interest to readers of PRSB.

Appendix A

In this paper, the authors present the hypothesis that a hybridisation event can lead to an adaptive radiation, and that subsequent “fission and fusion” between or among the descendants of that adaptive radiation can provide the genetic material necessary for future adaptive radiations. This would explain why we see repeated bursts of speciation in the same lineages in nature. This is a hypothesis that the authors and their colleagues have developed elsewhere, but to their knowledge and mine, it has not been illustrated with a proof-of-concept model. Here, the authors present that proof-of-concept model, and much of it is clear and convincing. I think this is an exciting manuscript, and I believe it would be an excellent fit in Proceedings B.

That said, there are a couple of places where either the authors’ results do not completely support their conclusion, or perhaps where I don’t completely understand the results and so would benefit from more explanation:

Major points:

1. The mechanism the authors propose has two parts. First, a hybridisation event provides the genetic diversity to initiate an adaptive radiation. Because of the hybrid origin of the radiation, the descendant lineages host very different sets of alleles. Second, the fission of lineages protects this diversity, because each lineage maintains its differentiated set of alleles (i.e., that is, there is no homogenisation, as we would expect in a panmictic population). As a result, a subsequent fusion of lineages can recreate the diversity needed for a new adaptive radiation. To demonstrate this mechanism, the authors must show that both the first and second parts above are necessary to the process.

For spatially repeated AR, I think they have done this. If only one source species is introduced into region 1, there is divergence and local adaptation in that region. However, when the descendant lineages reach region 2, they do not have enough genetic diversity to start another radiation. So, initial hybridisation is necessary. If regions 1 and 2 are connected by a single corridor with one niche, or a single corridor with two niches and weak selection so the lineages in the corridor are not sufficiently isolated (i.e., there is no fission), then there is (often) not enough diversity to start another radiation in region 2. So, fission is necessary (or at least it helps).

Interestingly, if there is a single corridor with two habitats and moderate or strong divergent selection, then it appears that the initial hybridisation event is *not* needed. In this case, negative frequency-dependent disruptive selection in the corridor maintains enough genetic diversity in the corridor populations to start a new radiation in region 2. So, while the authors have shown that their mechanism can promote repeated adaptive radiations, I think they have also shown that it is not *necessary* for repeated adaptive radiations. If I am correct about this, I think the authors should mention this in the discussion. If I am wrong, I think they should explain why I am wrong.

For temporally repeated AR, I am less convinced about the authors' mechanism. I think the authors have shown that fission is necessary. If the refugia in the refugial phase are isolated, there is often enough remaining genetic variation to start a new radiation after the refugial phase. If they are not sufficiently isolated, there is not. But, could this happen even if the initial radiation were started by a single species rather than by hybridisation? I don't see where the authors have tested this, and without this evidence I do not think they have fully demonstrated the mechanism they propose.

2. The authors state in several places (starting in the abstract line 20-21) that "single spatially confined hybridization events can promote many episodes of adaptive radiation." But, I am not sure that is what we see. Rather, we see an initial hybridization event leading to an initial adaptive radiation, and then genetic diversity maintained in isolated lineages. Then, we see *another* hybridization event, this time between descendant lineages. This is especially true if we believe that the descendant lineages are "species" as adaptive radiation implies. The authors have called this second hybridization "fusion," perhaps to indicate that it is somehow different from hybridization, but I don't understand how it is different. To be clear, I don't think the result becomes any less interesting or exciting if there are multiple hybridizations. But, I think the authors should either explain how fusion differs from hybridization, or remove (or clarify) the claim that a single hybridization event is responsible for multiple radiations.

I also have several minor objections or suggestions. As a reviewer, I don't believe I have the right to "insist" on anything. It is the authors' paper, not mine. But, objections are things I really think the authors should fix or clarify. Suggestions are just places where I think they could make the paper better, but if they ignore them I would not expect a response. I have marked suggestions so the authors can tell the difference.

Line 17-19: <suggestion> I found this sentence a bit cumbersome and hard to follow.

Line 44: Could you give (approximate) generations instead of years, since these will be more relevant for evolution?

Line 52-54: <suggestion> I found the explanation in lines 49-52 clear and easy to follow, but the analogy in lines 52-54 very confusing. I would cut it.

Line 94: Should the summation start with $k=1$?

Lines 99-100: It was not clear to me from this line whether crossovers could split the derived nucleotides of a given allele or if they always occurred between alleles.

Line 106: I recommend using different subscripts to indicate patches and individuals. (I especially would not use the same subscript to indicate different things in the same expression!)

Line 107: I would use different symbols to indicate the set of derived loci and the number of individuals in a habitat patch. (Currently they are both M).

Line 111: Should be "... selects a single ..."

Line 118: In this line, the authors claim that ecological differentiation leads to speciation in their model, but I am not sure I believe it. There is no assortative mating. The authors claim that there is hybrid inviability, but I don't know if I believe that either. There are no Dobzhansky-Muller incompatibilities. Where there is reduced fitness of hybrids, it is simply because the optimal environment for those hybrids does not exist (for example, in corridors with H₁ and H₃), and sometimes that habitat comes into existence later. Originally I had listed the lack of well-defined speciation in a model that purports to be about adaptive radiation among my major concerns, but for the most part the authors have carefully avoided using the term "species" to refer to differentiated lineages in their model. I would go through and make sure they have avoided it everywhere. (Alternatively, they could clearly define what they mean by species, and demonstrate that the lineages meet this definition. But, that seems like a lot of work, and it might not be very convincing anyway!)

Line 159-161: I was not sure if this means 30 total simulations per parameter set or 900.

Line 216, "Such a situation": I was unsure what situation this referred to. I think it is the case where there is divergence after hybridization?

Line 264-267: I was initially worried about the lack of selection in the refugia, since intuitively I would expect strong selection in refugia. These lines deal with that in part. But now, I wonder if the results would be different if the habitat in the refugia were extreme (e.g., H₁ or H₅, or even a new H₀ or H₆) than intermediate. I would expect that in nature refugia are often extreme habitats. If the authors have to run more simulations for something else, I might check this. If not, I would at least mention it in the discussion.

Line 279: "episodes" should be "episode."

Lines 287-288, "... our simulations are most likely underestimating ..": I might say "... this assumption is conservative with regard to ..." After all, there are other assumptions in the model that almost certainly favour the authors' mechanism. I think the biggest one is likely to be the assumption of a potentially infinite number of strictly additive alleles. This means that there are an infinite number of ways to get to exactly the ecological phenotype, and that different solutions can be recombined without penalty. This doesn't bother me in a proof-of-concept model, but it does make me doubt whether the model really underestimates the difficulty/frequency with which this mechanism can produce repeated adaptive radiations in nature. I wonder this especially for the case of refugia with similar selective regimes. I am not suggesting the authors should investigate this. I don't think that is necessary in a proof-of-concept model. But, I do think it is worth mentioning in the discussion that this assumption might facilitate the mechanism the authors propose.

Line 481: I think figure 1 would benefit from a more extensive caption. In particular, I would like to be told what the blue bands and yellow arrows mean. I can probably guess, but I would rather not have to!

Lines 498-500, “that the number of species in region 2 was proportional to (i) trait range in region 2 and (ii) the number of genotypically distinct clusters of individuals that are found only in region 2 (Fig. S3).”: I don’t understand (even when I am looking at figure S3).

Appendix B

Prof. Dr. Daniel Costa

Editor

Proceedings of the Royal Society B: Biological Sciences

Dear Prof. Costa,

Please find attached our manuscript (MS) which was originally entitled “Fission and fusion of lineages can promote nested adaptive radiations within clades of hybrid origin” (RSPB-2020-0941) in the first submission. We have now thoroughly revised our manuscript based on the comments from reviewers and editor. We found their comments very helpful, and have taken advantage of utilizing them as much as possible in revising the MS.

Specifically, our revision includes a thorough rewriting of the abstract section and two paragraphs of the background section as well as the change of the title in order to address major comments by reviewers. We believe that these revisions provide an appropriate framing of the novelty in our study. We have also carried out additional simulations following reviewers’ comments, results of which strengthen the support for our conclusions. Detailed point-by-point responses to all points raised by the reviewers follow below. For convenience, reviewers’ comments are numbered and shown in red. We believe that our responses address all the points raised by the reviewers.

Although the revision required additional discussion and clarifications, we were able to keep the length of MS the same as the original version (6682 words, including the title page, abstract, key words, references, table and figure legends; 5 figures, and no tables in the main text). We achieved this by condensing the text.

This work has not been published elsewhere, nor is it under consideration for publication elsewhere.

We hope that the revised MS will be deemed suitable for publication in *Proceedings of the Royal Society B: Biological Sciences*. Thank you very much for your time to consider our manuscript.

Sincerely yours,

Kotaro Kagawa and Ole Seehausen

Graduate School of Life Sciences, Tohoku University, 2-1-1 Katahira, Aoba, Sendai, Miyagi, 980–8578 Japan.

E-mail: kagawakoutarou@gmail.com

TEL: +81 80 6643 6880

Responses to Reviewer #1

Reviewer #1 was generally positive but had “one general concern about the framing of the novelty of the paper, and several specific concerns that are all minor, although one does suggest an additional set of simulations”.

We below describe our responses to each comment.

[R1-1A] To me the novelty of this study is fairly subtle, and is not sufficiently clear from the abstract. There is much recent work (cited in this study) showing that combining standing adaptive variation can facilitate speciation and adaptive radiation. The process underlying recurrent radiations in the present paper is, in my opinion, not a distinct process. Just like the first radiation, the second radiation is triggered by bringing together old adaptive variation. I realise that the variation has already been through an initial hybridisation event, but I don't think that fundamentally changes the nature of the second radiation. Therefore, the main findings of this study are the mechanisms by which sufficient adaptive variation can be maintained: essentially population structure either by physical isolation or heterogeneous habitat. **[R1-2]** I worry that using the term “dynamic fission and fusion”, rather than the familiar terms above, might artificially inflate the distinctiveness of the mechanisms presented.

To be clear, I do still think this is a valuable study. It has certainly helped me better understand the circumstances under which pre-existing variation may spawn multiple radiations. **[R1-3]** There are also some nice nuanced points that could receive more emphasis: (1) The fact that the trait is polygenic and there is a distribution of fitness effects means that even isolated lineages under the same selective regime will maintain distinct variation. (2) A second radiation can fail to occur in a new habitat if the colonising lineages expand at different rates.

[R1-1B] In summary, I think the paper (especially the abstract) would be improved if the authors were more specific and up front about the fact that this paper is not presenting an entirely new evolutionary process, it is exploring circumstances under which the pre-existing variation can be maintained and recombined, such that the same process can be repeated.

[Response to R1-1A, B]

We appreciate this comment. While what we aimed to explain in the previous MS was basically not different from what suggested by the reviewer, the comment made us realize that our previous MS, especially its abstract, was misleading. Especially, we totally agree that our previous abstract was misleading about the novelty of the study due to too much emphasis on the broad implication of our findings and too much abstracted and shortened explanations on our scientific question, findings, and its novelty. We also agree that the term “dynamic fission and fusion” could be misleading (see [Responses to R1-2]; as we explained there, we did not aim to imply that the “fusion” of sub-lineages promoting the second adaptive radiation is an essentially different process from hybridization promoting adaptive radiation). To address these concerns, we have thoroughly rewritten the abstract to convey more specific and accurate explanation on our question,

findings, and its novelty as much as possible. Specifically, the revised abstract clarifies the following points:

- (1) The novel finding from our simulations are the mechanisms for the long-term maintenance and spatial spread of ancient admixture variation. We are proposing mechanisms that can extend the effect of one major hybridization event (e.g. two divergent lineages merging upon secondary contact) from facilitating one adaptive radiation to promoting many adaptive radiations in episodes that are distinct in space and time (lines 21-24).
- (2) It is true that secondary admixture between populations that emerged from a hybrid population (“fusion” in the previous MS) generates genetic variation same as the ancestral hybridization event, but the ultimate origin of the genetic variation generated by hybridization of sub-lineages is the older ancestral hybridization event. We clarified this point by providing more detailed explanation of the process by which admixture between sub-lineages that emerged from one hybrid population can restore old genetic variation from the initial hybridization event (lines 24-27). Key here is that the necessary, yet difficult and rare event - the admixture of two old evolutionary lineages - only needs to happen once. Yet, if the resulting hybrid lineage dynamically splits and reunites (through speciation and its reversal or through spatial isolation and its removal), the massive effect of the one unlikely event can be propagated through space and time (lines 17-24).

Similarly, we have extensively revised the Background section to convey better explain that the paper is not presenting an entirely new process and how our study extends previous theories. Specifically, the revised Background explicitly describes the following points:

- (1) The present paper is proposing novel mechanism for long-term maintenance and spatial spread of adaptive genetic variation, which can extend the (previously known) radiation facilitating effect of hybridization to be able to facilitate recurrent adaptive radiation (lines 70-77).
- (2) The aim of our simulations is “to theoretically clarify geographic and ecological conditions under which genetic variation generated by single hybridization events can persist for long periods and spread to distant areas, facilitating repeated occurrence of adaptive radiation.” (lines 98-100)
- (3) Our hypothetical mechanism for the maintenance of genetic variation is an extension of classical population genetics theories. In the previous MS, we explained this point in the Discussion section. To be more explicit about which part of our hypothesis is theoretically novel, the revised MS explains how our hypothesis extends classical population genetics theories in the Background section (lines 79-88).
- (4) The process by which secondary admixture of sub-lineages restore old genetic variation is same as the original hybridization event creating genetic variation. We made this clearer by adding more detailed explanation on this process (lines 88-91).
- (5) Hybridization promoting adaptive radiation is highly contingency-dependent and unlikely to occur repeatedly in the same lineage. Thus, this hybridization seems unlikely to explain recurrent adaptive radiation of the same lineage (lines 40-46). A key implication from the hypothetical scenario that our simulations considered is that “the rare and unlikely event of hybridization between lineages of the right genetic distance may only need to occur once for promoting recurrent adaptive radiations in the hybrid lineage” (lines 55-56).

[Response to R1-2]

We did not expect that the term “fission and fusion” is misleading since this term has been used in several previous relevant papers to express temporal isolation of lineages followed by secondary admixture between them which is driven by ecological, geographic, and climatic mechanisms (for example, Grant BR & Grant PR. 2008 “Fission and fusion of Darwin’s finches populations.” *Philos. Trans. R. Soc. B Biol. Sci.*). In the previous MS, we used this term to briefly summarize our hypothetical mechanism for the maintenance of high evolvability generated by an ancestral hybridization event: that is, a temporal isolation of sub-lineages maintaining genetic variation and subsequent secondary admixture between them reestablishing standing genetic variation within population. However, thanks to comments from both reviewer #1 and #2 (R2-3), we noticed that the meaning of the term is not necessarily well established and the term could be a source of misunderstandings. Therefore, we have revised the MS to avoid using this term. Specifically, we made the following revisions:

- (1) We changed the title to “The propagation of admixture-derived adaptive radiation potential” (the original title was “Fission and fusion of lineages can promote nested adaptive radiations within clades of hybrid origin”).
- (2) In the revised MS, we used terms like “...the hybrid population becomes separated into isolated sub-lineages. Subsequent secondary hybridization...” (lines 22-23), “subdivision of the hybrid population into genetically isolated lineages” (lines 77-78), “subsequent secondary admixture” (line 78; 88), “temporal isolation and subsequent secondary admixture” (lines 378), “speciation and geographic isolation and subsequent secondary admixture of species or sub-lineages” (lines 384-385) rather than “fission and fusion”, to avoid the confusions caused by terminology.

[Response to R1-3]

Following the suggestion, we made two revisions. First, we emphasized that the polygenic control of ecological traits is a key assumption for the maintenance of adaptive variation in both introduction and discussion section (lines 85-88; 332-333). Second, we made a separate paragraph discussing about empirical implications of our results that spatially repeated AR via two isolated corridors could fail when range expansion of one of two parental lineages precedes the hybridization between them (lines 354-358).

[R1-4] 1. Lines 319-329 argue that the process seen here is distinct from the transporter hypothesis, because “secondary admixture of sub-lineages in a place distant from the original hybridisation episode generated a wide range of phenotypes, including those with low fitness in the first radiation”. I agree that the process here is distinct from the transporter hypothesis, because the genotypic clusters that emerged in the second radiation tended to be distinct from those in the first radiation, but I do not see what part of the results shows that the second radiation involved phenotypes that would have low fitness in the first radiation. Surely to test that idea one would require the second radiation to have a distinct set of optima? I think including such a model would be a nice addition to the paper.

[Response to R1-4]

Thank you for the helpful comment. While the sentence aimed to indicate the observation that a wide range of phenotypes arose in a short period of time immediately after the secondary admixture between two sub-lineages, we did not confirm that such phenotypic variations can indeed facilitate adaptive radiation into novel ecological niches that do not exist in the first adaptive radiation. Therefore, we conducted two lines of additional simulations. First, we performed a simulation in which different sets of fitness optima were assumed for region 1 and 2 (Fig. S9). This simulation confirmed that secondary admixture of sub-lineages in the region 2 can promote adaptive radiation into novel habitats, of which optimal traits are different from those in the first radiation. Second, we constructed a model with two ecological traits and simulated a situation that only the first trait is subject to divergent selection in the region 1 and only the second trait is subject to divergent selection in the region 2 (Fig. S10). Simulations with this model again confirmed that secondary admixture between sub-lineages in the region 2 could promote adaptive radiation in the region 2, despite that the second trait is subject to stabilizing natural selection in both region 1 and corridors. We have argued that these results demonstrate that secondary admixture of sub-lineages can promote adaptive radiation into novel environments that do not exist in the first radiation (lines 250-253; 331).

[R1-5] 2. The very first sentence of the Results (line 186) is quite long and difficult to follow, and I also found it a bit misleading. It is not clear on what is meant by long-distance, nor does it account for the fact that expansion along a single corridor can lead to a second AR, provided the corridor has a heterogeneous environment. My first thought was that the entire sentence is unnecessary, but if you would like to start with a summary, I suggest a more general statement that the occurrence of a second AR depended on the maintenance of variation during expansion, which could occur in a few different ways.

[Response to R1-5]

Thank you for the suggestion. We have removed the sentence (line 204).

[R1-6] 3. Line 99: “crossover recombination, which occurs at a rate r per base-pair in meiosis, creates chromosomes with novel combinations of derived nucleotides”. Although more detail is provided in the Appendix, I think it is necessary to mention here that recombination can occur within loci (i.e. to separate derived mutations at the same locus), as this is not obvious from the paragraph as it is currently written.

[Response to R1-6]

In the revised manuscript, we wrote in the main text that recombination can occur within loci (lines 117-118).

[R1-7] 4. Appendix S1: “Every derived nucleotide has a unique randomly assigned position within its target locus”. Does this mean it is an infinite sites mutation model, such that recurrent mutation of a single site is not possible? Is this reasonable given the time scales and mutation rates being considered?

[Response to R1-7]

Yes, we implemented the simulation as an infinite site mutation model. We revised the sentence to make this fact clear (line 29 of the SI).

We think that this approximation does not affect our conclusion. While multiple mutations on exactly the same position of the genome cannot happen in the infinite sites mutation model, our model explicitly considering physical linkage structure can simulate very similar situations: if multiple mutations occur on positions very close to each other, recombination between them is extremely unlikely. More essential difference between our model and the real genome is in the possible number of states for each nucleotide. In the real genome, every nucleotide can have only 4 states and thus backward mutations are possible, whereas the infinite sites mutation model does not incorporate this constraint. Thus, in very long time scales where nonnegligible fraction of nucleotides experience multiple substitutions, the infinite sites mutation model will be an inaccurate approximation due to the lack of backward mutations. Nonetheless, multiple substitutions on exactly same nucleotides must be extremely rare in the time scale of our simulations. Namely, the number of mutations fixed in each parental lineage after the allopatric evolution for 200000 generations was about 700 (Fig. S1), which is much smaller compared to the total length of adaptation-relevant loci, 2×10^6 bps (Note: although our simulation model treats positions in the genome as continuous rather than discrete values, length of genetic regions can be converted to base-pair length because we defined recombination distance using a realistic recombination rate per base-pair by assuming that the whole genome is 3×10^9 bps in length and each of adaptation-relevant loci is 5000 bps long). Moreover, we think that the lack of backward mutations in our simulations is not going to matter as much for our study aiming to provide a qualitative proof-of-concept based on a simplified model of genetics.

[R1-8] 5. Figure S1 I think it might be clearer to replace “mutations accumulated” with “mutations fixed” or simply “substitutions”.

[Response to R1-8]

Following the suggestion, we have replaced “mutations accumulated” with “mutations fixed” (line 395 of the SI).

[R1-9] 6. Figure S2 legend explains why an initial radiation does not produce many species when selection is weak, but it does not explain why very strong selection also fails to produce an initial radiation following hybridisation. Is this because some habitats are never colonised?

[Response to R1-9]

Yes, very strong selection hindered adaptive radiation because it made invasion of new habitats difficult. This result was owing to our model assumption that locally maladaptive phenotypes reduces survival rate even in

the absence of competitors. In the revised manuscript, we explained the reason why strong selection failed to produce an adaptive radiation (lines 413-420 of the SI).

[R1-10] 7. Figure 2 (and others like it) is too small when drawn across a single pdf page.

[Response to R1-10]

For Figs. 2, 4, S11 (of the previous MS), we made each panel bigger by changing the layout of panels and cutting some less informative panels which was showing magnification of evolutionary dynamics in short terms. As for the Fig. S5, we made the figure bigger by rotating the page 90 degrees.

[R1-11] 8. Effect of breaking linkage structures (lines 271-269). To me, comparing Figures 5 and S12, it is very difficult to make out whether or not there is a difference. Could this be described quantitatively?

[Response to R1-11]

Thank you for the reasonable comment. We agree that it was difficult to find the differences between Figs. 5, S9, and S12 (of the previous MS). In the revised MS, results from both simulations with and without free recombination, which were previously shown in separate figures, are shown together in the Fig. S11. Additionally, we provided a statistical test for the difference between results with and without free recombination for one parameter set with which the difference is pronounced. There were statistically significant differences in the trait range and the number of genotypic clusters in the renewed radiation, although we did not find significant difference in the number of species. This was probably because the difference between results with and without free recombination was pronounced when natural selection was weak, but with weak selection speciation was basically unlikely (with this parameter value, the number of species in the renewed radiation was mostly 1). We described results of statistical tests in the figure legend (lines 583-589 of the SI).

[R1-12] 9. Lines 289-293 argue that, in addition to standing adaptive variation, new mutations that arise in the separated lineages would further facilitate future AR. I partly agree, but I think one exception is cases where the temporary isolation involves complete breakdown of selection. The key aspect of the older standing variation is that it arose in a heterogeneous environment with selection. Mutations that fix in the absence of selection will surely not facilitate future AR, otherwise we would expect any increase in variation to facilitate future AR, such as increasing the population size, right?

[Response to R1-12]

Although we are not totally sure if we are correctly understanding the comment, at least in our simulation model, mutation arising in each isolated sub-lineage in the absence of natural selection can contribute to phenotypic variation generated by subsequent hybridization between sub-lineages. The below figure shows a result of simulation confirming this. In this simulation, we first simulated allopatric evolution of two lineages

in the absence of natural selection (left panels). In this period, the ecological trait of both lineages stochastically changed to random directions due to genetic drift causing fixation of mutations affecting the trait. This process led to fixation of many mutations in each lineage, of which phenotypic effects were often compensating each other. Mutations fixed independently in each lineage can give rise to large phenotypic variation when genomes of two lineages are recombined in the hybrid population (transgressive segregation). The simulation of hybridization between two lineages in the absence of natural selection (the right panel) confirmed this effect as phenotypic variation in the hybrid population exceeded phenotypic ranges of both parental lineages combined (the effect is most visible in the first 100 generations after the hybridization). If this hybrid population encounter an ecological opportunity where the ecological trait x becomes subject to divergent natural selection, the elevated phenotypic variation will facilitate adaptive radiation.

[R1-13] 10. Line 347: it would be helpful to add some information on whether there is evidence that either multiple corridors or heterogeneous corridors existed in these empirical cases.

[Response to R1-13]

We added a sentence referring available evidence that multiple corridors existed in the case of the Lake Victoria region (lines 348-350). Additionally, we briefly argued about possible ways to test whether multiple corridors or heterogeneous corridors existed in general cases of spatially repeated adaptive radiation (lines 350-352).

[R1-14] 11. Line 367: This should refer to Appendix S5.

[Response to R1-14]

We appreciate this comment pointing out the mistake. We revised this sentence as suggested (line 378).

Responses to Reviewer #2

Reviewer #2 valued our study providing a proof-of-concept on our hypothetical mechanism by which hybridization-induced genetic variation can promote multiple episodes of adaptive radiation, but also concerned that “there are a couple of places where either the authors’ results do not completely support their conclusion, or perhaps where I don’t completely understand the results and so would benefit from more explanation”. Additionally, the reviewer raised several minor objections and made some helpful suggestions to improve the MS. We below describe our responses to each comment.

Major points:

[R2-1] 1. The mechanism the authors propose has two parts. First, a hybridisation event provides the genetic diversity to initiate an adaptive radiation. Because of the hybrid origin of the radiation, the descendant lineages host very different sets of alleles. Second, the fission of lineages protects this diversity, because each lineage maintains its differentiated set of alleles (i.e., that is, there is no homogenisation, as we would expect in a panmictic population). As a result, a subsequent fusion of lineages can recreate the diversity needed for a new adaptive radiation. To demonstrate this mechanism, the authors must show that both the first and second parts above are necessary to the process.

For spatially repeated AR, I think they have done this. If only one source species is introduced into region 1, there is divergence and local adaptation in that region. However, when the descendant lineages reach region 2, they do not have enough genetic diversity to start another radiation. So, initial hybridisation is necessary. If regions 1 and 2 are connected by a single corridor with one niche, or a single corridor with two niches and weak selection so the lineages in the corridor are not sufficiently isolated (i.e., there is no fission), then there is (often) not enough diversity to start another radiation in region 2. So, fission is necessary (or at least it helps).

Interestingly, if there is a single corridor with two habitats and moderate or strong divergent selection, then it appears that the initial hybridisation event is *not* needed. In this case, negative frequency-dependent disruptive selection in the corridor maintains enough genetic diversity in the corridor populations to start a new radiation in region 2. So, while the authors have shown that their mechanism can promote repeated adaptive radiations, I think they have also shown that it is not *necessary* for repeated adaptive radiations. If I am correct about this, I think the authors should mention this in the discussion. If I am wrong, I think they should explain why I am wrong.

For temporally repeated AR, I am less convinced about the authors’ mechanism. I think the authors have

shown that fission is necessary. If the refugia in the refugial phase are isolated, there is often enough remaining genetic variation to start a new radiation after the refugial phase. If they are not sufficiently isolated, there is not. But, could this happen even if the initial radiation were started by a single species rather than by hybridisation? I don't see where the authors have tested this, and without this evidence I do not think they have fully demonstrated the mechanism they propose.

[Response to R2-1]

We think that these concerns are owing to a misunderstanding on the logical structure of our paper, which could have occurred because our writing was not fully clear. First of all, what we conclude from our simulation results is that: genetic variation generated through an ancestral hybridization event and the subsequent isolation and secondary admixture (or “fission and fusion”) of sub-lineages *can promote* (but not necessarily *necessary for*) recurrent adaptive radiation. To obtain this conclusion, we performed simulations in which the source of genetic variation is limited to the initial ancestral hybridization event. That is, we did not allow spontaneous mutations after the initial hybridization event, and we excluded the effect of standing genetic variation within each parental population (for this purpose, we simulated hybridization between two clonal populations each composed of a single haploid genome randomly sampled from one of two parental populations). Therefore, by model assumptions, the initial hybridization event is necessary for both the first and the second adaptive radiation in both simulations of spatially and temporally repeated AR scenarios (without the initial hybridization, evolution cannot occur in our simulations). Owing to these model assumptions, if recurrent adaptive radiation could occur in our simulations, we can safely conclude that genetic variation generated by the initial hybridization event can promote recurrent adaptive radiation. Our simulations revealed that both spatially and temporally repeated adaptive radiation are feasible but need certain conditions that induce temporal isolation and subsequent admixture (“fission and fusion”) of sub-lineages. These results provide a proof-of-concept for our hypothetical mechanism by which genetic variation from the ancestral hybridization event can promote recurrent adaptive radiation, because contamination of other sources of genetic variation was excluded from our simulations. In the revised MS, we explicitly explained this logical structure of our simulation study in both Method and Discussion sections (lines 153-159; lines 301-304).

In our understanding, the reviewer #2 suggested providing the comparison between simulations with and without the initial hybridization event in the presence of spontaneous mutations. Although we had thought of this simulation experimental design at first, we did not adopt it because the design cannot safely demonstrate our hypothetical mechanism. That is, in the presence of spontaneous mutations after the initial hybridization event, genetic variation generated by the secondary admixture between sub-lineages contains not only the old variation from the initial hybridization event but also new variation generated through recombination of de novo mutations that arose after the isolation of sub-lineages and had fixed in either sub-lineage. Therefore, with spontaneous mutations, the secondary admixture of sub-lineages promoting a secondary episode of adaptive radiation may be owing to not only maintenance of old but also creation of new genetic variation.

Therefore, we adopted simulations without spontaneous mutations. As we have explained above, this simulation experimental design does not need the comparison of simulations with and without the initial hybridization to test if hybridization-induced genetic variation can promote recurrent adaptive radiation. Another reason that we did not compare simulations with and without the initial hybridization in the main part of our MS was simply because it was outside the scope of the present study. A previous simulation study by one of the authors (Kagawa, K. & Takimoto, G. 2018 Ecol. Lett.) provided the comparison between simulations with and without hybridization, and their results demonstrated that hybridization can promote (non-recurrent) adaptive radiation. In the present study, we were interested in whether and how this radiation facilitating effect of hybridization can be maintained for long-term and spread to distant places, promoting multiple episodes of adaptive radiation. Nonetheless, our Fig. S2 provides the comparison between simulations with and without hybridization in the presence of spontaneous mutations to reconfirm that hybridization can promote single episode of (i.e. non-recurrent) adaptive radiation in the model of the present study. The results confirmed that rapid adaptive radiation was extremely unlikely without hybridization-induced genetic variation under parameter conditions that we used.

In agreement with reviewer's comments, we do think that recurrent adaptive radiation does not always need the initial hybridization event if spontaneous mutations are allowed, as we explicitly maintained in the revised MS (lines 302-303). This was the reason that our conclusion was that genetic variation generated through hybridization *can promote* (instead of "*is necessary for*") recurrent adaptive radiation. As shown by results of Fig. S2 and a previous theoretical study (Kagawa, K. & Takimoto, G. 2018 Ecol. Lett.), adaptive radiation can occur without hybridization by spontaneous mutations alone under some specific parameter conditions, while adaptive radiation was much more likely with hybridization in wide range of parameter conditions including our default parameter set. Thus, both spatially and temporally repeated adaptive radiation should be feasible without hybridization under some specific conditions where adaptive radiation could occur with only spontaneous mutations. Accordingly, our simulations do not generally conclude that the initial hybridization is necessary for recurrent adaptive radiation. However, we note that recurrent adaptive radiation by spontaneous mutations alone cannot explain the empirical example of cichlid radiations in the Lake Victoria region. Genomic evidence suggests that old genetic variation generated by a single episode of ancient hybridization had promoted multiple episodes of adaptive radiations in multiple distant lakes of this region, at least one of which had started long after the hybridization event. To explain this example, we need mechanisms for long-term maintenance and spatial spread of hybridization-induced genetic variation, and we suggest that temporal isolation and subsequent admixture ("fission and fusion") of sub-lineages is a theoretically feasible candidate mechanism.

[R2-2] 2. The authors state in several places (starting in the abstract line 20-21) that "single spatially confined hybridization events can promote many episodes of adaptive radiation." But, I am not sure that is what we see. Rather, we see an initial hybridization event leading to an initial adaptive radiation, and then genetic diversity maintained in isolated lineages. Then, we see *another* hybridization event, this time between descendant

lineages. This is especially true if we believe that the descendant lineages are “species” as adaptive radiation implies. The authors have called this second hybridization “fusion,” perhaps to indicate that it is somehow different from hybridization, but I don’t understand how it is different. To be clear, I don’t think the result becomes any less interesting or exciting if there are multiple hybridizations. But, I think the authors should either explain how fusion differs from hybridization, or remove (or clarify) the claim that a single hybridization event is responsible for multiple radiations.

[Response to R2-2]

We appreciate the comment, with which we realized that our writing was misleading. We have two answers to the comment. First, as we have described in the [Response to R1-2], we were not aiming to imply that “fusion” is a something essentially different from the hybridization. As we noticed that there may not be a consensus on what this term means and that the term could have been a source of misunderstandings, we revised the entire manuscript to avoid using this term (Response to R1-2). Second, by saying “single spatially confined hybridization events can promote many episodes of adaptive radiation” (or similar phrases), we meant that genetic variation generated by single spatially confined hybridization events between genetically distant lineages can promote many episodes of adaptive radiation (as we described in the [Response to R2-1], the initial hybridization event was the ultimate source of genetic variation for both the first and second adaptive radiation in our simulations). We totally agree that our expression was confusing, given that secondary admixture of sub-lineages (or “fusion”) is also a hybridization event. To solve this problem, we replaced phrases like “single spatially confined hybridization events can promote...” with the following phrases: “...exceptional genetic variation, once generated through a rare and unlikely hybridization event, can facilitate...” (line 25-26), “genetic variation once generated by a singular hybridization event may...” (line 47), “genetic variation generated through a single ancient hybridization episode has promoted” (line 49), “...exceptional genetic variation once generated by a singular event of hybridization between distant evolutionary lineages can...” (lines 75-76), “...genetic variation generated by single hybridization events can...” (lines 98-99), “...genetic variation generated by a singular geographically localized hybridization episode can promote...” (lines 301-302), “...genetic variation generated by single episodes of hybridization, spatially narrowly confined, can affect...” (lines 387-388).

[R2-3] Line 17-19: <suggestion> I found this sentence a bit cumbersome and hard to follow.

[Response to R2-3]

Thank you for the suggestion. To address this and other comment (R1-1), we have revised the entire abstract, including this sentence, to make it more specific and better understandable (lines 15-27).

[R2-4] Line 44: Could you give (approximate) generations instead of years, since these will be more relevant for evolution?

[Response to R2-4]

Following the suggestion, we provided an approximate number of generations for cichlids corresponding to 100,000 years (line 53).

[R2-5] Line 52-54: <suggestion> I found the explanation in lines 49-52 clear and easy to follow, but the analogy in lines 52-54 very confusing. I would cut it.

[Response to R2-5]

Lines 52-54 (of the previous MS) aimed to explain why natural selection inhibit not only the long-term maintenance but also spatial spread of hybridization-induced high adaptive genetic variation. We have revised the sentence to improve the readability (lines 63-67).

[R2-6] Line 94: Should the summation start with $k=1$?

[Response to R2-6]

We have corrected the equation as suggested (line 112). Thank you for the helpful comment.

[R2-7] Lines 99-100: It was not clear to me from this line whether crossovers could split the derived nucleotides of a given allele or if they always occurred between alleles.

[Response to R2-7]

Thank you for the comment. Reviewer #1 also made the same point (R1-6), and we have revised the sentence (line 117) as we have explained in the [Response R1-6].

[R2-8] Line 106: I recommend using different subscripts to indicate patches and individuals. (I especially would not use the same subscript to indicate different things in the same expression!)

[Response to R2-8]

We agree that our symbol was confusing. As we have introduced x_{opt_H} ($H = H_{1,2,...5}$) in the preceding sentence, we revised the sentence using the symbol H instead of H_i as follows: “The growth performance P_i of an individual i in a habitat H is given by $exp\{-\varphi(x_i - x_{opt_H})^2\}$, where...” (lines 123-124).

[R2-9] Line 107: I would different symbols to indicate the set of derived loci and the number of individuals in a habitat patch. (Currently they are both M).

[Response to R2-9]

Following the suggestion, we changed the symbol to indicate the set of derived loci from M to S (lines 114; 116).

[R2-10] Line 111: Should be "... selects a single ..."

[Response to R2-10]

Thank you, and we followed the suggestion (lines 129-130).

[R2-11] Line 118: In this line, the authors claim that ecological differentiation leads to speciation in their model, but I am not sure I believe it. There is no assortative mating. The authors claim that there is hybrid inviability, but I don't know if I believe that either. There are no Dobzhansky-Muller incompatibilities. Where there is reduced fitness of hybrids, it is simply because the optimal environment for those hybrids does not exist (for example, in corridors with H₁ and H₃), and sometimes that habitat comes into existence later. Originally I had listed the lack of well-defined speciation in a model that purports to be about adaptive radiation among my major concerns, but for the most part the authors have carefully avoided using the term "species" to refer to differentiated lineages in their model. I would go through and make sure they have avoided it everywhere. (Alternatively, they could clearly define what they mean by species, and demonstrate that the lineages meet this definition. But, that seems like a lot of work, and it might not be very convincing anyway!)

[Response to R2-11]

Thanks to the comment, we noticed that our usage of the term species was not accurate. Below, we first explain what we meant by the term "species" and then we describe our response to this issue.

We are aware of that definitions of "species" and "speciation" are controversial, especially in cases where crossing between "species" can produce fertile hybrids, and this was the reason that we avoided to call sub-lineages species in most parts of the manuscript. The "species" that we defined in the Appendix S4 is based on clustering of ecologically similar individuals and the observed level of gene flow among clusters. Since our model did not incorporate mechanisms for permanent reproductive isolation such as Dobzhansky-Muller incompatibilities or behavioral isolation, "species" of our model (1) can produce fully fertile hybrids and (2) will merge if spatial isolation between habitats and divergent natural selection are removed. Thus, we agree that our definition of "species" does not meet the Mayer's biological species concept in the strict sense. However, in our understanding, "speciation" of our model does meet the definition of incipient ecological speciation with immigrant inviability (Nosil 2012). That is, with ecological differentiation between populations, reduced ecological fitness of immigrants in their non-native environments prevents them to survive and reproduce, thereby excluding gene flow among populations. Additionally, in our model, ecological natural selection reduces fitness of inter-"specific" F1 hybrids because their phenotypes fall into fitness valleys. This mechanism, which is called extrinsic hybrid inviability, also contributes to reduce gene flow between ecologically differentiated populations. With these mechanisms, ecologically differentiated "species" of our model showed low levels of inter-"specific" gene flow (Appendix S4) and thus, they had

separate gene pools that will evolve independently.

We agree that the “species” that we defined in the Appendix S4 does not indicate good species with permanent reproductive isolation. Thus, we replaced “species” and “ecological speciation” with “incipient species” (lines 136; 238; 239; 243; 262; 498) and “incipient ecological speciation” (lines 135). Additionally, the revised manuscript clarifies that incipient species of our model lacks permanent reproductive isolation and can merge if spatial isolation between habitats or divergent natural selection are removed (lines 136-137; lines 164-171 of the SI).

[R2-12] Line 159-161: I was not sure if this means 30 total simulations per parameter set or 900.

[Response to R2-12]

We meant that we ran only 30 simulations in total for allopatric evolution of parental lineages pairs (with the default parameter set). The same 30 pairs of parental lineages were used to perform 30 replications of simulation of hybridization and the following evolutionary dynamics for all parameter sets that we examined. In other words, we examined the likelihood of recurrent adaptive radiation with varying parameter values, while not varying the set of initial conditions. To make this clearer, we revised the sentence as follows: “For this analysis, we first simulated evolution of 30 independent pairs of two parental lineages with the default parameter set (Table S1). Then, for all parameter sets, we performed 30 simulations of hybridization and evolution that follows using the same set of 30 parental lineage pairs to form the initial hybrid populations.” (line 179-181).

[R2-13] Line 216, “Such a situation”: I was unsure what situation this referred to. I think it is the case where there is divergence after hybridization?

[Response to R2-13]

We rewrote the sentence corresponding to the line 216 of the previous MS to reduce the uncertainty. To address the comment R1-3, we have moved the sentence to a new a paragraph in the Discussion section (lines 356-357).

[R2-14] Line 264-267: I was initially worried about the lack of selection in the refugia, since intuitively I would expect strong selection in refugia. These lines deal with that in part. But now, I wonder if the results would be different if the habitat in the refugia were extreme (e.g., H_1 or H_5, or even a new H_0 or H_6) than intermediate. I would expect that in nature refugia are often extreme habitats. If the authors have to run more simulations for something else, I might check this. If not, I would at least mention it in the discussion.

[Response to R2-14]

Following the suggestion by the reviewer, we conducted simulations with a scenario that the environmental

condition during the refugial phase exerts natural selection favoring an extreme phenotype, $x = 10$ (Fig. S14). Simulation results supported that the isolation between three refugia can maintain genetic variation for the reestablishment of the adaptive radiation even if the habitat in refugia is extreme. However, results of simulation replications under systematically varied parameter values showed that reestablishment of specialists for habitats H_4 and H_5 was much less likely to occur compared to the case with intermediate refugial habitat. This is due to the increased distance between the phenotype of refugial sub-lineages and the fitness optima of H_4 and H_5 in the second adaptive radiation.

[R2-15] Line 279: “episodes” should be “episode.”

[Response to R2-15]

Thank you for the comment. During the revision, the sentence had been removed from the manuscript.

[R2-16] Lines 287-288, “... our simulations are most likely underestimating ..”: I might say “... this assumption is conservative with regard to ...” After all, there are other assumptions in the model that almost certainly favour the authors’ mechanism. I think the biggest one is likely to be the assumption of a potentially infinite number of strictly additive alleles. This means that there are an infinite number of ways to get to exactly the ecological phenotype, and that different solutions can be recombined without penalty. This doesn’t bother me in a proof-of-concept model, but it does make me doubt whether the model really underestimates the difficulty/frequency with which this mechanism can produce repeated adaptive radiations in nature. I wonder this especially for the case of refugia with similar selective regimes. I am not suggesting the authors should investigate this. I don’t think that is necessary in a proof-of-concept model. But, I do think it is worth mentioning in the discussion that this assumption might facilitate the mechanism the authors propose.

[Response to R2-16]

Thank you for the helpful suggestion. Following the suggestion, we revised the sentence as: “In this aspect, our simulations are conservative regarding the effects of hybridization between sub-lineages in promoting recurrent radiation.” (lines 307-308). Additionally, the revised manuscript emphasizes that polygenic control of the ecological trait is the key assumption for the mechanism that we propose (lines 332-333).

[R2-17] Line 481: I think figure 1 would benefit from a more extensive caption. In particular, I would like to be told what the blue bands and yellow arrows mean. I can probably guess, but I would rather not have to!

[Response to R2-17]

We agree this comment. However, at the same time, such a caption will be highly redundant with the Method section and the length restriction makes it difficult to provide thorough explanation on the details of two scenarios in the caption. Thus, in the revised MS, we explained the meanings of blue bands and blue squares in the figure caption (lines 485-486) and added the following sentence to the caption: “Details of each

scenario are in the Methods.” (line 487). The meaning of yellow arrows is explained in the Method section (line 168).

[R2-18] Lines 498-500, “that the number of species in region 2 was proportional to (i) trait range in region 2 and (ii) the number of genotypically distinct clusters of individuals that are found only in region 2 (Fig. S3).”: I don’t understand (even when I am looking at figure S3).

[Response to R2-18]

Thank you for the comment. We revised the sentence as follows: “Other metrics of evolutionary diversification gave consistent results (Fig. S3).” (lines 499-500).